# Neutron-upscattering enhancement of the triple-alpha process

J. Bishop [1✉], C. E. Parker [1], G. V. Rogachev [1,2,3], S. Ahn [1,11], E. Koshchiy [1], K. Brandenburg[4], C. R. Brune [4], R. J. Charity [5], J. Derkin[4], N. Dronchi[6], G. Hamad[4], Y. Jones-Alberty[4], Tz. Kokalova [7], T. N. Massey[4], Z. Meisel[4], E. V. Ohstrom[6], S. N. Paneru [4], E. C. Pollacco[8], M. Saxena[4], N. Singh[4], R. Smith [9], L. G. Sobotka [5,6,10], D. Soltesz[4], S. K. Subedi[4], A. V. Voinov [4], J. Warren[4] & C. Wheldon [7]

The neutron inelastic scattering of carbon-12, populating the Hoyle state, is a reaction of interest for the triple-alpha process. The inverse process (neutron upscattering) can enhance the Hoyle state's decay rate to the bound states of $^{12}C$, effectively increasing the overall triple-alpha reaction rate. The cross section of this reaction is impossible to measure experimentally but has been determined here at astrophysically-relevant energies using detailed balance. Using a highly-collimated monoenergetic beam, here we measure neutrons incident on the Texas Active Target Time Projection Chamber (TexAT TPC) filled with $CO_2$ gas, we measure the $3\alpha$-particles (arising from the decay of the Hoyle state following inelastic scattering) and a cross section is extracted. Here we show the neutron-upscattering enhancement is observed to be much smaller than previously expected. The importance of the neutron-upscattering enhancement may therefore not be significant aside from in very particular astrophysical sites (e.g. neutron star mergers).

[1] Cyclotron Institute, Texas A&M University, College Station, TX 77843, USA. [2] Department of Physics & Astronomy, Texas A&M University, College Station, TX 77843, USA. [3] Nuclear Solutions Institute, Texas A&M University, College Station, TX 77843, USA. [4] Edwards Accelerator Laboratory, Department of Physics & Astronomy, Ohio University, Athens, OH 45701, USA. [5] Department of Chemistry, Washington University, St. Louis, MO 63130, USA. [6] Department of Physics, Washington University, St. Louis, MO 63130, USA. [7] School of Physics and Astronomy, University of Birmingham, Birmingham, UK. [8] IRFU, CEA, Université Paris-Saclay, Gif-Sur-Yvette, France. [9] Department of Engineering and Mathematics, Sheffield Hallam University, Sheffield S1 1WB, UK. [10] McDonnell Center for the Space Sciences, Washington University, St. Louis, MO 63130, USA. [11]Present address: Center for Exotic Nuclear Studies, Institute for Basic Science (IBS), Daejeon 34126, Republic of Korea. ✉email: jackbishop@tamu.edu

The triple-alpha process is extremely important in nucleosynthesis. Three $\alpha$-particles produce $^{12}$C in a multi-step process via the intermediate $^8$Be and through the Hoyle state[1] which resonantly enhances the reaction rate by several orders of magnitude. The rate for generation of bound $^{12}$C depends on the radiative-decay (sequential gamma-decay and pair-production) width of the Hoyle state as the alternative is $\alpha$-particle decay that will regenerate the starting material and force the process to start over again. The dominant contribution to the radiative-width is sequential gamma-decay via the $2_1^+$ state at 4.44 MeV. It has long been suggested that in certain stellar environments, additional decay components run in parallel to the radiative-width and could greatly increase the production of bound $^{12}$C[2]. The additional decay components, known as upscattering (previously referred to as particle-induced de-excitation of nuclei), occurs when light particles (neutron, protons and $\alpha$ particles) interact with the resonant Hoyle state and carry away the excitation energy as kinetic energy, allowing the nucleus to de-excite as portrayed in Fig. 1. The presence of the Coulomb barrier means that for low temperatures, contributions from neutrons are expected to dominate. The importance of neutron upscattering to the triple-alpha reaction rate has previously been estimated[3]. This previous study relied on Hauser-Feshbach calculations for the $^{12}$C$(0_2^+)(n_2, n_{0,1})$ cross sections, the neutron subscript denotes the excited state in the $^{12}$C system.

This work presents the measurement of the $^{12}$C$(n, n_2)3\alpha$ cross section (in the relevant energy region) allowing, with a combined R-matrix fit, for the cross-sections for the astrophysically-relevant inverse processes to be determined. The results show the influence of neutron-upscattering is much less important than previously thought.

## Results

**Neutron beam**. To perform this measurement, an intense, fast, and quasi-monoenergetic neutron beam was required which was provided by the Edwards Accelerator Laboratory at Ohio University[4]. A direct-current deuteron beam of 1–2 μA from the 4.5-MV tandem accelerator was incident on a deuterium-filled gas cell of length 7.98 cm held at 760 Torr. The gas cell entrance window was a 2.5-μm-thick Havar foil and the beam was stopped by a gold foil at the end of the gas cell. The $d(d, n)$ reaction is forward-focused and collimated via a multi-material system placed in the 30-m time-of-flight tunnel with an aperture

diameter of 0.75 cm at a distance of 2 m from the gas cell. Highly-collimated fast neutron beams with ten energy steps of energies from 7.2 to 10.0 MeV, with full widths ranging from ~ 350 to 250 keV, were generated in this fashion with a typical intensity of 5000 neutrons per second.

Rough beam normalisation was provided by charge integration of the deuteron beam in the gas cell as well as counting neutrons from a Stilbene detector placed 1-m away from the gas cell and an NE-213 detector placed 30 m from the gas cell at the end of the time-of-flight tunnel.

**TexAT time projection chamber**. The Texas Active Target (TexAT) Time Projection Chamber (TPC)[5] was used to measure the $^{12}$C$(n, n_2)3\alpha$ cross section. The sensitive volume of the TPC is $224 \times 245 \times 130$ mm$^3$. The TexAT TPC was filled with 50.0 ($\pm$0.2) Torr of $CO_2$ gas which was refreshed at a rate of 50 sccm. The Thick Gas Electron Multiplier[6] were combined with Micromegas for an overall gas gain of 5000. The field cage voltage provided an electric field of 69 V cm$^{-1}$, drifting ionised electrons with a velocity of 0.75 cm μs$^{-1}$. For this experiment, in addition to the TPC region of TexAT, four 625-μm-thick Si detectors were placed $\approx$50 cm from the beam entrance. These silicon detectors provided overall high-precision beam normalisation via the number of detected proton recoils from a 30.0-μm-thick $CH_2$ foil placed inside of TexAT via the $^1$H$(n, p)$ elastic scattering reaction. The thickness of this foil and its variation over its surface is known to roughly 3%. The entrance to the TexAT chamber is an aluminium flange of thickness 2.87 mm which corresponds to a loss of neutron flux of 3%. TexAT was placed inside the Edwards Accelerator Laboratory time-of-flight-tunnel behind the collimator system and was at a distance of 4 m from the gas cell to minimise the neutron beam spot size whilst maintaining exceptional neutron background suppression.

With TexAT, 3D tracks of the charged particles can be reconstructed along with information about the energy deposition. Multiple tracks can be resolved with a separation of $\approx$2 mm. An example track can be found in Supplementary Fig. 1 along with other additional information as part of the Supplementary information.

**$^{12}$C$(n, \alpha_0)$/$^{16}$O$(n, \alpha_{0,1})$ cross section**. In order to identify $(n, \alpha)$ interactions, events with a Micromegas-pixel hit multiplicity >20 (to eliminate elastic-scattering recoil tracks) had their tracks fitted with three different reactions: $^{12}$C$(n, \alpha_0)$ (leaving $^9$Be in the ground state), $^{16}$O$(n, \alpha_0)$ (leaving $^{13}$C in the ground state) and $^{16}$O$(n, \alpha_1)$ (leaving $^{13}$C in the first-excited state). A $\chi^2$ value for each of the three reaction channels was calculated. For each, the overall track fit (sum of the distances of the track points to the fitted track lines), deviation of the angle of the products from those expected, and difference in range of the heavy particle that stops inside the TPC region from those expected from the two-body kinematics contributed to the $\chi^2$. As well as integrated cross sections, the statistics for several energy settings were sufficient to generate complete coverage angular distributions, which will be reported in a future publication.

**$^{12}$C$(n, n_2)3\alpha$ cross section**. To extract the inelastic scattering contribution to the Hoyle state, the RANSAC-fitting technique[7] was used on the 3D tracks to identify events with three $\alpha$-particles in the exit channel of which the desired events are a subset. More information on the RANSAC technique is presented in Supplementary Fig. 2. The separation of the desired inelastic events (leading to the Hoyle state) from those that result from $\alpha$ decay of $^{13}$C populating excited states of $^9$Be that further alpha decay is

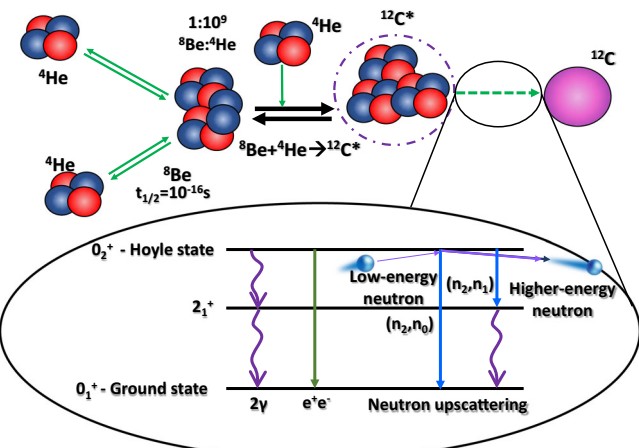

**Fig. 1 Overview schematic diagram.** This details the steps involved in the triple-alpha process, showing the contribution of neutron upscattering to the decay width to bound states. A low-energy neutron interacts with the Hoyle state leaving carbon-12 in the ground-state (or first-excited state) and the extra energy is carried away with the neutron.

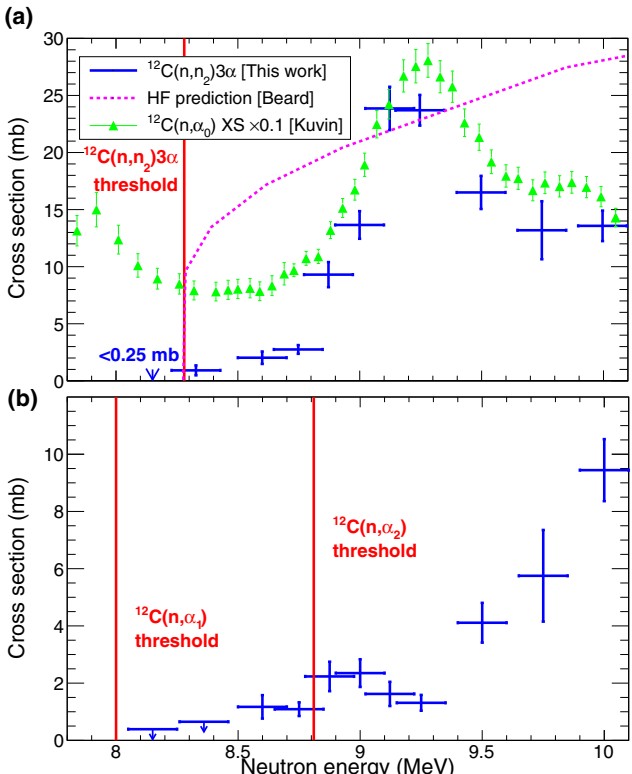

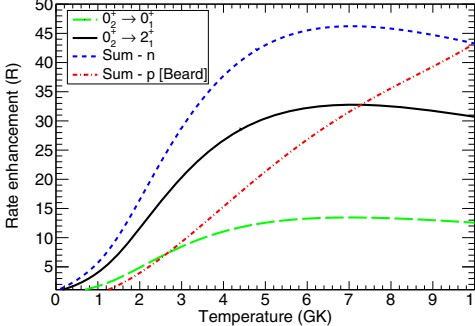

**Fig. 3 Enhancement of the Hoyle radiative-width via neutron upscattering as a function of temperature.** Contributions from upscattering to the $^{12}$C ground state (green dashed) and to the $^{12}$C first-excited state (black solid). The sum of these is shown in dotted blue for a neutron density of $10^6$ g cm$^{-3}$. The comparable contribution from protons as calculated in ref. [3] is shown by the dot-dashed red line.

$^{12}$**C($n$, $\alpha_{1,2}$).** As well as the $3\alpha$ final state arising from the inelastic scattering to the Hoyle state, one may also populate excited states in $^9$Be that are unbound, therefore yielding an identical final state of $3\alpha + n$. The kinematics of this reaction is very different, whereby one track is much longer than the other two, and it is kinematically possible for this higher energy $\alpha$-particle to have a scattering angle >90°. More details on the channel separation is available in the Supplementary Figs. 3, 4. It is important to separate this cross section as it does not contribute to the neutron-induced upscattering in stellar environments. Instead, this cross section forms three helium nuclei per interaction and, therefore, provides a more-sizeable contribution (versus the $^{12}$C($n$, $\alpha_0$)) as a gas-forming reaction in high-neutron flux environments such as fusion reactors blanket materials. The cross sections are shown in Fig. 2b, where the separation between $\alpha_{1,2}$ (with $\alpha_1$ and $\alpha_2$ meaning the $^9$Be is left in the first- or second-excited state respectively) is not established here. This reaction can clearly be seen to proceed through a resonance in $^{13}$C a few hundred keV lower than the resonance for the $^{12}$C($n$, $n_2$) peak.

**Neutron-upscattering rate enhancements.** Taking the experimentally-measured cross sections, an R-Matrix fit and the detailed-balance principle (see Methods for more information) were employed and the neutron-upscattering rate enhancements were calculated. For a nominal neutron density of $10^6$ g cm$^{-3}$, the enhancements for both the $^{12}$C($0_2^+$)($n_2$, $n_1$) and $^{12}$C($0_2^+$)($n_2$, $n_0$) are shown in Fig. 3 as a function of temperature. The total rate enhancement (dotted blue line in Fig. 3), $R$, can be approximated by the functional form:

$$R = A \tanh(BT_9^3 + CT_9^2 + DT + E) + \frac{FT_9}{1 + \exp\left(\frac{T_9 - G}{H}\right)},  \quad (1)$$

with the coefficients $A = 36.6$, $B = 0.486$, $C = 0.0775$, $D = 0.130$, $E = 0.0374$, $F = -5.09$, $G = 2.32$ and $H = 1.85$.

For a neutron density of $10^6$ g cm$^{-3}$, the enhancement can be as high as 46 at 7 GK, smaller than the factor of >100 originally predicted from Hauser-Feshbach calculations[3]. At higher temperatures (or larger electron fractions), the proton upscattering contribution will dominate and can have a significant impact on nucleosynthesis[9].

## Discussion

The cross section for the inelastic neutron scattering of $^{12}$C to the Hoyle state was experimentally measured at astrophysically-relevant energies. Using the TexAT TPC, the three $\alpha$-particles

**Fig. 2 Cross sections as a function of neutron energy for channels of interest. a** $^{12}$C($n$, $n_2$)$3\alpha$ cross section (blue error bars) measured across the energy range of astrophysical interest with the Hauser-Feshbach (HF) predictions from ref. [3] overlaid in dashed magenta. The $^{12}$C($n$, $\alpha_0$) cross section, scaled by a factor of 0.1 is overlaid with green triangles with data from ref. [8] (Table III) to show the similarity in form with the experimental data from this work. The cross section below the threshold was determined to be < 0.25 mb at the 95% C.L. where zero Hoyle events were observed. **b** $^{12}$C($n$, $\alpha_{1,2}$)$^9$Be$^\star$ cross section with the two thresholds for the centroid of the $^9$Be$^\star$ states marked by solid red lines. The lowest two energy points had zero observed events so the upper limit at the 95% C.L. is shown. For both plots, the x-axis error bars represent the total width of the neutron energy spectrum rather than its standard deviation due to the non-Gaussian nature of the beam energy profile. The y-axis shows purely statistical error bars at the 1σ level.

presented in Supplementary Figs. 3, 4. The data for the latter are discussed in Sec. III E.

Events with bad fits were visually examined to eliminate any deficiencies of the RANSAC technique or track reconstruction. There was also a subset of events where two of the $\alpha$-particle tracks from the decay of $^8$Be were insufficiently spatially separated; therefore the 3D track only showed two tracks. Where these events could not be attributed to ($n$, $\alpha$) reactions and the nature of this event could not unambiguously be determined, this contributed to the overall uncertainty for the cross section.

The $^{12}$C($n$, $n_2$)$3\alpha$ cross section over the neutron energy range of interest is shown in Fig. 2a. It is immediately apparent that, contrary to the predictions of Hauser-Feshbach calculations previously performed for this reaction[3], the cross section does not abruptly rise near the threshold and the overall magnitude of the cross section is smaller than expected. This difference has a profound effect for small temperatures where the energies near threshold are most important. The structure of the cross section is similar to that for $^{12}$C($n$, $\alpha$) (shown by green triangles in Fig. 2a), strongly suggesting the importance of one or more $^{13}$C compound-nucleus states around 13.5 MeV.

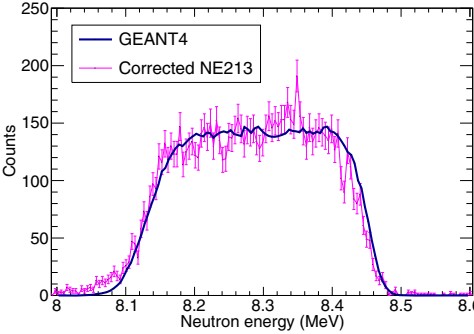

**Fig. 4 Incident neutron energy spectrum.** The spectrum is shown from the NE-213 detector placed at 30 m with a pulsed beam (magenta points) with corrections for the interaction location within the gas cell applied. The neutron energy spectrum from the 7.98-cm gas cell as predicted by GEANT4 is overlaid (solid blue line) and normalised to reproduce the amplitude of the flat region between 8.2 and 8.36 MeV. Error bars correspond to a statistical uncertainty of $1\sigma$.

from the decay of the Hoyle state were measured and the contributions from the $^{12}C(n, \alpha)^9Be^{\star}$ reaction were separated. The cross section is dominated by a small number of resonances in $^{13}C$. With a multi-channel R-Matrix fit, the cross section for the $^{12}C(2_1^+)(n_1, n_2)3\alpha$ reaction was also extracted which cannot be measured directly experimentally.

The importance of neutron scattering to the triple-alpha process has been a longstanding question for many decades. Using detailed balance, the current experimental results show that the measured cross sections are significantly suppressed near the threshold in comparison to Hauser-Feshbach predictions. At these low temperatures, where previously the neutron enhancement factor was predicted to be greater than 100, the enhancement is instead small, of the order of unity. Given the suppression of the cross section around the threshold, the importance of this neutron scattering process is not as high as originally thought[3]. Therefore, this resolves the question of the importance of neutron and proton upscattering (the latter of which was shown to be highly-important[9]) at lower temperatures where Coulomb repulsion heavily suppresses the proton upscattering contribution. The neutron upscattering enhancement may be more significant for the extremely low electron fractions found in neutron star mergers.

This measurement demonstrates the use of neutron-induced reactions with an active-target TPC and demonstrates the applicability of this technique for similar measurements. This technique will be applied to understanding the role of additional neutron upscattering reactions such as in the so-called Holy Grail $^{12}C(\alpha, \gamma)^{16}O$ reaction where the 9.585 MeV $1^-$ state radiative decay width is heavily suppressed due to an isospin-forbidden E1 transition. Any small additional neutron upscattering contribution to the many lower-lying states may provide a sizeable enhancement in a stellar environment where reactions such as $^{13}C(\alpha, n)$ generate many neutrons.

## Methods

**Beam characterisation**. In order to characterise the beam, a pulsed neutron beam was generated and a time-of-flight spectrum was recorded with an NE-213 placed at a distance of 30 m from the gas cell. For this measurement, the incident deuteron energy was 5.36 MeV and the gas cell pressure was a slightly higher 988 Torr. A correction was made to account for the 7.98-cm-long gas cell (which produced a distance uncertainty) via extracting an initial energy assuming the interaction occurred at the middle of the gas cell. A more accurate distance was then determined due to the fact the energy of the neutron produced is linearly related to the interaction depth of the deuteron inside the gas cell and a more accurate neutron energy was then extracted. With this correction, the E-TOF measurement agreed well with the expected neutron spectrum from the $d(d, n)$ reaction as simulated in

GEANT4[10], as long as we accounted for a small ~0.03% air contaminant in the calibration runs. This agreement is shown in Fig. 4.

For the runs where a direct current beam was used, the neutron energy spectrum was validated using the energy spectrum of the $^1H(n,p)$ scattering events in the Si detectors. Reproducing the simulated width expected in the Si detectors not only validated the incident neutron-energy spectrum but also the thickness of the $CH_2$ foil which contributes to a large degree via the energy loss associated with the interaction depth in the foil.

**Normalisation**. The principle technique used was $^1H(n, p)$ scattering. The associated uncertainties from this were the thicknesses of the foil and counting statistics. The conversion from silicon counts to neutron flux was calculated using GEANT4 for each energy. The beam normalisation was confirmed by neutron flux estimates from the integrated deuteron beam current, which were in agreement with the deuteron differential cross section uncertainties at zero degrees.

**Detailed balance enhancement**. Given the experimentally-measured reaction rates, the inverse astrophysically-relevant reaction rate can be determined using detailed balance and the enhancement this has on the radiative width of the Hoyle state evaluated. Following the prescription detailed in[3], the additional rate factor from upscattering, R, (where R = 1 implies the upscattering enhancement is equal to the unenhanced electromagnetic radiative width, therefore doubling the triple-alpha reaction rate) is given by:

$$R = k_n \rho_n T_9^{-1.5}(2J_i + 1) \times \int_0^\infty \sigma_{nn'}(E)(E - Q) \exp(-11.605E/T_9) dE, \quad (2)$$

where $k_n$ is a factor ($6.787 \times 10^{-6}$ K$^\frac{3}{2}$cm$^3$g$^{-1}$ for inelastic neutron scattering for this system, using the radiative width suggested in ref.[1]), $\rho_n$ is the neutron density in g cm$^{-3}$, E is the energy in MeV above the threshold in the c.m., $T_9$ is the temperature in GK, $\sigma_{nn'}(E)$ is the inelastic scattering cross section in mb, Q is the reaction Q value in MeV, and $J_i$ is the angular momentum of the final state in the astrophysical case.

**R-Matrix fit**. The contribution of neutron upscattering to the $2_1^+$ state in $^{12}C$ is particularly important, in part due to the $(2J_i + 1)$ factor that increases the importance of this channel by a factor of 5. To extract the $^{12}C(2_1^+)(n_1, n_2)3\alpha$ cross section, a multi-channel R-Matrix fit was performed[11] using data from $^{12}C(n, n_0)$ and $^{12}C(n, n_1)$[12], $^{12}C(n, \alpha_0)$[8], $^9Be(\alpha, n_2)$[13], as well as our current experimental data for the $^{12}C(n, n_2)3\alpha$ channel. The values for the partial widths $\Gamma_{n0}$, $\Gamma_{n1}$ and $\Gamma_{n2}$ were therefore constrained. These fits (Supplementary Figs. 5–8 and Supplementary Table 1) indicate the importance of levels at 13.28, 13.28, 13.57, and 13.76 MeV with $J^\pi = \frac{1}{2}^+, \frac{3}{2}^-, \frac{7}{2}^-$, and $\frac{5}{2}^+$, respectively. Apart from the first level, there are states at these energies with known, or suspected, spin-parities[14]. There is evidence for another state near 13.0 MeV but with an unknown spin[14]. A spin-parity assignment of $J^\pi = \frac{1}{2}^+$ for this state generates a large partial decay width to the Hoyle state, which allows for a good fit to our measured excitation function. A $J^\pi = \frac{1}{2}^+$ state in this energy region, with the clustered structure of $^{12}C(0_2^+) \otimes s_\frac{1}{2}$, has been predicted from antisymmetrised molecular dynamics calculations[15].

## Data availability

The experimental data from this study are available upon reasonable request by contacting the primary author. Due to the large size of the data, they cannot be hosted publicly.

## Code availability

The analysis code used to analyse these data are available at https://github.com/jackbishop-phys/neutron_upscatter_TexAT.

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

## Acknowledgements

This work was supported by the U.S. Department of Energy, Office of Science, Office of Nuclear Science (under awards no. DE-FG02-93ER40773 (J.B., G.V.R., S.A., E.K.), DE-FG02-88ER40387 (K.B., C.R.B., G.H., Y.J.A., T.N.M., Z.M., S.P., M.S., N.S., D.S, S.S., A.V.V., J.W.), DE-SC0019042 (Z.M., M.S., N.S., D.S.), DE-NA0003883 (C.R.B., Y.J.A., T.N.M., S.P., J.W.), DE-NA0003909 (K.B., Z.M., A.V.V.), DE-FG02-87ER-40316 (R.J.C., N.D. and L.G.S.), by National Nuclear Security Administration through the Center for Excellence in Nuclear Training and University-Based Research (CENTAUR) under grant number DE-NA0003841 (J.B., C.E.P., G.V.R., S.A., E.K., N.D., E.V.O., L.G.S.) and by the UK STFC Network+ Award Grant number ST/N00244X/1 (Tz.K., R.S., and C.W.). This work also benefited from support by the National Science Foundation under Grant No. PHY-1430152 (JINA Center for the Evolution of the Elements). G.V.R. also acknowledges the support of the Nuclear Solutions Institute.

## Author contributions

J.B., C.E.P., G.V.R., S.A., E.K., C.R.B., T.N.M., Z.M., E.C.P., L.G.S., R.S., Tz.K., and C.W. provided planning for the work contained in this paper which G.V.R. and L.G.S. originally conceived. J.B., C.E.P., G.V.R., S.A., E.K., and E.C.P. developed the TexAT TPC used. J.B., C.E.P., G.V.R., S.A., E.K., K.B., C.R.B., R.J.C., J.D., N.D., G.H., Y.J.A., T.N.M., Z.M., E.V.O., S.N.P., M.S., N.S., L.G.S., D.S., S.K.S., A.V.V., and J.W. participated in the experiment in person. The analysis and primary manuscript preparation were performed by J.B. with S.A. providing assistance with the simulation work verification. G.V.R. and L.G.S. were the PIs for the main grant for this experiment (CENTAUR). All co-authors contributed to the final manuscript.

## Competing interests

The authors declare no competing interests.
