## [Peer Review File · Nature Communications]

Neutron-upscattering enhancement of the triple-alpha processREVIEWER COMMENTS

Reviewer #1 (Remarks to the Author):

The present paper aims at examining the enhancement of the triple-alpha reaction rate by the neutron-induced de-excitation of the Hoyle [$^{12}\text{C}(0+2)$] state. In normal stellar environments, the ^{12}C nuclei are mainly synthesized by the radiative decay of the Hoyle state, but it has been suggested that the ^{12}C production rate in high-density environments might be significantly enhanced by particle-induced de-excitation processes, i.e. exothermic inelastic scattering between the Hoyle state and background particles such as protons, neutrons, and alpha particles. It has been expected that neutrons dominantly contribute to the enhancement because the neutron-induced de-excitation is not hindered by the Coulomb barrier. However, the cross section for the neutron-induced de-excitation of the Hoyle state at astrophysical energies has been unknown. The authors reported the cross section for the neutron-induced de-excitation of the Hoyle state to the ground state [$^{12}\text{C}(0+2)(n2,n0) ^{12}\text{C}(0+1)$] for the first time by measuring the time-reverse reaction [$^{12}\text{C}(0+1)(n0,n2)^{12}\text{C}(0+2)$] with a novel active-target time-projection chamber (ATTPC). The authors also estimated the enhancement of the ^{12}C production rate due to the neutron-induced de-excitation to the first excited state [$^{12}\text{C}(0+2)(n2,n1)^{12}\text{C}(2+1)$] by the R-matrix analysis, and concluded that the enhancement of the ^{12}C production rate due to the neutron-induced de-excitation is not larger than that expectation in the previous work [3].

The enhancement of the ^{12}C production rate due to background particles in high-density environments have been uncertain for several decades because the experimental determination of the cross section for the $^{12}\text{C}(0+1)(n0,n2)^{12}\text{C}(0+2)$ reaction was not easy. The present work would be a remarkable achievement to improve our understanding of the triple-alpha reaction and various astrophysical phenomena if the result was reliable. However, the paper does not fully convince me to warrant the reliability of their measurement and analysis. I think that the authors should clarify the following remarks to publish their result in Nature Communications.

1. The authors should describe their analysis of the ATTPC data in more detail. I guess that the 3D tracking of the 3-body decay of the Hoyle state is not so straightforward. The tracking efficiency should depend on scattering angles, scattering points, and neutron energies. The authors should describe how they estimated the tracking efficiency, how much their tracking efficiency is, and how it depends on the scattering angles, points, and energies.

2. The correctness of the estimated efficiency is very important because it essentially determines the cross section and the ^{12}C production rate. Although the 3-body tracking is much more difficult than the 2-body tracking, the authors should demonstrate the correctness of the tracking efficiency by quantitatively comparing their 2-body $^{12}\text{C}(n0,\alpha0)^9\text{Be}$ cross section with the previous data from Ref.

[9] at least.

3. The event selection procedure is also important in the present analysis. The author claims that discrimination between the $^{12}\text{C}(0+1)(n0,n2)^{12}\text{C}(0+2)$ events from the $^{12}\text{C}(0+1)(n0,\alpha1)^9\text{Be}(1/2+1)$ and $^{12}\text{C}(0+1)(n0,\alpha2)^9\text{Be}(5/2-1)$ events is easy but I think it is not so trivial. The authors should show the invariant mass spectrum and the kinematical plot between the energy versus angle of the center of mass of the 3alpha particles to ensure their correct selection of the $^{12}\text{C}(0+1)(n0,n2)^{12}\text{C}(0+2)$ events.
4. The neutron-energy spectrum in Fig. 2 should be presented in wider energy range. I wonder how much background due to the Havar foil for the gas target were seen in high-energy region.
5. The authors should estimate and give the systematic uncertainty of the measured cross section because only the statistical uncertainty is given in Fig. 3.

Minor remarks:

1. The typical neutron flux on the ATTPC and the dimensions of the ATTPC should be given for readers' information.
2. In the legend of Fig. 3(a), " $^{12}\text{C}(n,\alpha2)3\alpha$ " should be corrected as " $^{12}\text{C}(n,n2)3\alpha$ ". " $^{12}\text{C}(n,\alpha)$ " should be " $^{12}\text{C}(n,\alpha0)^9\text{Be}$ " for consistency with Ref. [5] and for readers to understand.
3. It is better to show the neutron-energy range which is important to estimate the enhancement factors at various temperatures. If it is wider than that shown in Figs. 5, 6, 7, and 8, the authors should show the results from the R-matrix analysis in the wider energy range in those figures. If it is not wider, the authors should present the important energy range by something like gray shades in the figures.
4. The horizontal axes in Figs. 5, 6, 7, and 8 should be given as "Excitation energy of ^{13}C " in order not to confuse readers.
5. I guess that the $1/2+$ resonance at $E_x = 13.0$ MeV in ^{13}C was considered in the present R-matrix analysis but its parameters were not given in Table I. If any states other than $E_x = 13.28$, 13.57 , and 13.76 MeV in ^{13}C were considered in the R-matrix analysis, all those states should be tabulated with their widths in Table I.
6. The caption of Table I should be revised as "Excited states in ^{13}C include in the".

Reviewer #2 (Remarks to the Author):

This paper deals with the features of one component of the triple-alpha reaction, but they have done a good job of making a new measurement and then working out the consequences for the 3α reaction. The work all appears to be carefully and competently performed, though without the experimental data and apparatus details I cannot vouch for the numerical accuracy of the result.

My slightest correction is that the first legend in figure 3(a) should have n_2 , not α_2 .

That corrected, I am much in favour of it being published.

Reviewer #3 (Remarks to the Author):

Key results:

The rate, how fast carbon is formed in the centre of the stars, is one of the most important questions of nuclear astrophysics. Since the initial development of the concept in the early 1950s, the nuclear decays of the Hoyle state in C-12 at 7.6 MeV have attracted significant attention.

In the laboratory, the Hoyle state dominantly decays by alpha particle into Be-8. However, a small portion of the time, C-12 will instead disintegrate into three alpha particles. Stable carbon is formed through the emission of an E2 transition, leading to the first excited 2+ state, or by an electric monopole transition, decaying directly to the ground state of C-12. In an astrophysical environment, alternative decay processes involving upscattering of protons, alpha particles or neutrons can enhance the overall carbon production.

This paper is the first detailed experimental study to determine the enhancement of the triple-alpha reaction rate due to neutron-upscattering. Accurate determination of the total reaction rate has a large impact on our understanding of nucleosynthesis.

Methodology and validity:

The experiments have been carried out using highly collimated, quasi-monoenergetic neutron beams, covering the 8 to 10 MeV energy range, which is most relevant to astrophysical scenarios. A comprehensive experimental setup was used to monitor the beam and detect three alpha particles from

the decay of the Hoyle state. A key component of the setup was the Texas Active Target Time Projection Chamber. Care was taken to calibrate the spectrometers, allowing the precise reconstruction of events. Extensive Monte Carlo simulations have also been used.

A large data set was collected for inelastic neutron-scattering reactions, where the outgoing particle was an alpha particle or neutrons leading the final nuclear systems in excited or ground states.

The neutron-upscattering reaction rates have been extracted from a detailed balance of the measured 'normal' reaction rates. The experimental rates are compared with multi-channel R-matrix calculations. Details of the measured rates are given in Figures 5 to 8 in the Appendix. These figures demonstrate the high quality of the data.

To construct the plot of the enhancement of the radiative width of the Hoyle state due to neutron upscattering (Figure 4), the authors followed the procedure presented in ref [3] (Beard, Austin and Cyburt, 2017). This method was used by Davids and Boner (ApJ 166 [1971] 405) to study enhancement of the triple-alpha reaction by proton upscattering. The original formulation of the method was described by Fowler, Cauglan and Zimmerman (Ann. Rev. A.A. 5 [1967] 525). Like these earlier studies, the present paper describes the enhancement rate R according to equation (1) on page 4 of the manuscript. The detailed definition of the enhancement is not given in the paper, but can be obtained from ref [3]:

$R = \tau_{\gamma} / \tau_{n} = \tau_{\gamma} * N_n * \langle \sigma v \rangle_{n}$, where τ_{γ} is the radiative mean life of the Hoyle state, τ_{n} is the mean life of the neutron-upscattering channel, N_n is the neutron density and $\langle \sigma v \rangle_{n}$ is the cross section of the neutron upscattering. According to this definition, to determine R , one needs to use the value of the radiative lifetime of the Hoyle state. However, no details are given in the present paper as to what value has been used (the reviewer assumes it is included in the k_n factor).

The most recent values of the relevant quantities to determine the radiative width are given in the 2014 review by Freer and Fynbo, ref [1] of the manuscript. Using the adopted values from [1], one can obtain $\tau_{\gamma}=1.77(17)E-13$ s. It is very close to the value given in ref [3]: $\tau_{\gamma}=1.710E-13$ s. It should be noted that in the past few years, new results have been published on the E2 (Kibedi et al., PRL 125 [2020] 182701) and E0 (Eriksen et al., PRC 102 [2020] 024320) branching ratios, which change the value of τ_{γ} by about 34%: $\tau_{\gamma}=1.32(15)E-13$ s. Using the new value in determining the neutron-upscattering enhancement formula could further reduce the R values in Figure 4.

Details of the determination of k_n , as well as the radiative width used in the calculations of Figure 4 should be included. This will make the results more robust, allowing it to be used in model calculations.

Some minor comments and corrections:

(a) Figure 2: It is not clear how the GEANT4 calculations were normalised to the experimental points. It seems the calculated curve underestimates experiments.

(b) Figure 3 caption: The data of Kuvin, ref. [9] is drawn from their Table III and not the figure. Ref [9] also has error bars, typically 10%, and should be shown in the figure.

Summary:

Overall, it is a very impressive study, and, in my view, it should be accepted for publication in Nature Communications. However, I feel, that details of the parameterisation of R-ratio, outlined above, should be included in a minor revision of the paper.

Response to referees

We would first like to thank all of the referees for their comments and suggestions on the manuscript. Our responses to these queries are split by referee number below. The referee's original comments are in **red bold**, changes to the manuscript are highlighted in **blue** and general responses are left in black.

1 Response to referee #1

The present paper aims at examining the enhancement of the triple-alpha reaction rate by the neutron-induced de-excitation of the Hoyle [$^{12}\text{C}(0_2^+)$] state. In normal stellar environments, the ^{12}C nuclei are mainly synthesized by the radiative decay of the Hoyle state, but it has been suggested that the ^{12}C production rate in high-density environments might be significantly enhanced by particle-induced de-excitation processes, i.e. exothermic inelastic scattering between the Hoyle state and background particles such as protons, neutrons, and alpha particles. It has been expected that neutrons dominantly contribute to the enhancement because the neutron-induced de-excitation is not hindered by the Coulomb barrier. However, the cross section for the neutron-induced de-excitation of the Hoyle state at astrophysical energies has been unknown. The authors reported the cross section for the neutron-induced de-excitation of the Hoyle state to the ground state [$^{12}\text{C}(0_2^+)(n_2, n_0) ^{12}\text{C}(0_1^+)$] for the first time by measuring the time-reverse reaction [$^{12}\text{C}(0_1^+)(n_0, n_2) ^{12}\text{C}(0_2^+)$] with a novel active-target time-projection chamber (ATTPC). The authors also estimated the enhancement of the ^{12}C production rate due to the neutron-induced de-excitation to the first excited state [$^{12}\text{C}(0_2^+)(n_2, n_1) ^{12}\text{C}(2_1^+)$] by the R-matrix analysis, and concluded that the enhancement of the ^{12}C production rate due to the neutron-induced de-excitation is not larger than that expectation in the previous work [3].

The enhancement of the ^{12}C production rate due to background particles in high-density environments have been uncertain for several decades because the experimental determination of the cross section for the $^{12}\text{C}(0_1^+)(n_0, n_2) ^{12}\text{C}(0_2^+)$ reaction was not easy. The present work would be a remarkable achievement to improve our understanding of the triple-alpha reaction and various astrophysical phenomena if the result was reliable. However, the paper does not fully convince me to warrant the reliability of their measurement and analysis. I think that the authors should clarify the following remarks to publish their result in Nature Communications.

1. The authors should describe their analysis of the ATTPC data in more detail. I guess that the 3D tracking of the 3-body decay of the Hoyle state is not so straightforward. The tracking efficiency should depend on scattering angles, scattering points, and neutron energies. The authors should describe how they estimated the tracking efficiency, how much their tracking efficiency is, and how it depends on the scattering angles, points, and energies.

Here, we provide further detail on the tracking efficiency as a function of neutron energy, scattering point and scattering angle. One of these variables, the scattering point, is straightforward to explain. A cut (removing the first and last 42 mm of the active region) is made so that only interaction vertexes sufficiently inside the Micromegas sensitive region such that Hoyle events have enough active area for the decay α -particles to be reconstructed. The active region has a constant structure as a function of scattering point. To address the neutron energy and scattering angle dependence, simulations are shown here where a random scattering angle (isotropic in the centre-of-mass) is selected for the $^{12}\text{C}(n, n_2)3\alpha$ events which are reconstructed in 3D and then passed through the RANSAC process. $\approx 75\%$ of events were able to be automatically reconstructed with the RANSAC algorithm with the

remaining 25% of events displaying a situation where two α -particles cannot be disentangled due to their close proximity or, one α -particle track was too short to be picked up by the RANSAC technique. As discussed in the original manuscript, these events were manually examined and their incompatibility with any of the available (n, α) reactions was demonstrated and this subset of events contributed to the overall cross section uncertainty to remain conservative.

More details on the RANSAC fitting technique are provided here that are too in depth for the main manuscript. Firstly, a 3D point-cloud corresponding to the tracks inside the TPC is reconstructed from the waveforms and Micromegas which also encompasses the energy information at each point. More details about how this part of the analysis is performed are available in the previously-published TexAT NIM paper [1] as well as a more recent NIM paper looking at reconstructing the Hoyle state inside TexAT following β -decay [2]. Then, a custom-written RANSAC-based fitting routine is applied. In this method, three points in 3D space corresponding to three of the points in the point-cloud are randomly selected which correspond to points along the tracks from the three alpha-particles. For each randomly selected collection of three points, a chi-squared value is evaluated to analyse the quality of the RANSAC fit which is as follows:

$$\chi^2 = \sum_i^{N_{\text{points}}} \min_{(j=1,2,3)} d_{ij}^2, \quad (1)$$

where d_{ij} is the perpendicular distance for the i^{th} point in the point cloud to the line j which is generated from the interaction vertex to the randomly-selected three points for $j = 1, 2, 3$. The summation is over all N points in the point-cloud. The interaction vertex is selected by the position in the point-cloud that is the most upstream - as the three α -particles are forward-focused in the case of populating the Hoyle state. The number of random iterations is chosen such that there is a high probability to randomly select one point on each of the three arm tracks, for this work 10,000 iterations were chosen to avoid long analysis computing times.

The correctness of the estimated efficiency is very important because it essentially determines the cross section and the ^{12}C production rate. Although the 3-body tracking is much more difficult than the 2-body tracking, the authors should demonstrate the correctness of the tracking efficiency by quantitatively comparing their 2-body $^{12}\text{C}(n_0, \alpha_0)^9\text{Be}$ cross section with the previous data from Ref.[9] at least.

We agree that evaluating the $^{12}\text{C}(n, \alpha_0)^9\text{Be}$ is a good metric to evaluate the efficiency. Further work is currently underway to fully separate the $^{12}\text{C}(n, \alpha_0)$ and $^{16}\text{O}(n, \alpha_0)$ however preliminary cross section results are presented here which shows good agreement with the most-recent LANL results. Fig. 1 shows the preliminary cross sections for the $^{12}\text{C}(n, \alpha_0)$ where the large error bars are attributed to currently underway efforts to separate the $^{16}\text{O}(n, \alpha_1)$ events which have a similar Q-value and response inside the TPC making separation difficult. At certain energies where the $^{16}\text{O}(n, \alpha_1)$ contribution looks to be strong, this can cause large error bars. The channel overlap is also a function of the angular distribution for the two channels. The magnitude of the cross section is obviously well-reproduced here.

The event selection procedure is also important in the present analysis. The author claims that discrimination between the $^{12}\text{C}(0_1^+)(n_0, n_2)^{12}\text{C}(0_2^+)$ events from the $^{12}\text{C}(0_1^+)(n_0, \alpha_1)^9\text{Be}(1/2_1^+)$ and $^{12}\text{C}(0_1^+)(n_0, \alpha_2)^9\text{Be}(5/2_1^-)$ events is easy but I think it is not so trivial. The authors should show the invariant mass spectrum and the kinematical plot between the energy versus angle of the center of mass of the 3alpha particles to ensure their correct selection of the $^{12}\text{C}(0_1^+)(n_0, n_2)^{12}\text{C}(0_2^+)$ events.

The removal of $^{12}\text{C}(n, \alpha_{1,2})^9\text{Be}^*$ is certainly important to verify. As with the response to question 1, these reaction channels were simulated (for $E_n = 10$ MeV where the channel separation is most difficult) and passed into the RANSAC fitting technique and the invariant-mass calculated where the momentum vectors are determined by the range and direction of the track. The ^{12}C excitation energy distributions obtained for the Hoyle inelastic scattering and the $^{12}\text{C}(n, \alpha_1)^9\text{Be}$ and $^{12}\text{C}(n, \alpha_2)^9\text{Be}$ are shown in Fig. 2a. In addition, when manually inspecting the events that could not be well fitted with the RANSAC technique, the $^9\text{Be}^*$ events can be visually identified for events where one α -particle track is much longer than the others and has a large relative angle to the next-longest α -particle. This is shown and simulated in Fig. 3 where the dot product between the longest and second-longest α -particle is plotted against the length of the longest α -particle track for those events which don't have an obvious long α -particle which goes backwards (a clear signature that the event does not originate

Figure 1: Preliminary $^{12}\text{C}(n, \alpha_0)$ cross section from our data overlaid with the recent LANSCE result.

from the Hoyle state). This demonstrates the basis for visual inspection where the RANSAC fails to provide a good event and hence the invariant mass cannot be calculated. Therefore, one can well identify the Hoyle events against those from other channels.

For those events where the RANSAC was able to find three arms for the real data, the excitation function for the real data is also presented for those events classified as $^{12}\text{C}(n, n_2)3\alpha$ and $^{12}\text{C}(n, \alpha_{1,2})$ are shown separately in Fig. 2b. This shows the good separation between these different event types. Furthermore, from a holistic standpoint, the difference between the cross section as a function of energy for the $^{12}\text{C}(n, \alpha_{1,2})^9\text{Be}^*$ and the Hoyle suggests that any channel mismatching must be extremely small otherwise these two cross section plots would follow a similar behaviour. We have also placed Figs. 2 and 3 as a supplementary figure in the manuscript to clarify and visually quantify the event selection process. This is now also a separate document.

The neutron-energy spectrum in Fig. 2 should be presented in wider energy range. I wonder how much background due to the Havar foil for the gas target were seen in high-energy region.

A wider energy scale version of Fig. 2 in the manuscript is presented here in Fig. 4 where the absence of peaks at higher energies is apparent. The leading contribution from the Havar foil would originate from the cobalt which would give a peak around 12.6 MeV. One would expect lower energy neutrons to play a small part due to the deuteron breakup however even for the highest energies, these neutrons lie well below the $^{12}\text{C}(n, n_2)3\alpha$ threshold [3]. For the timing setup system used here, the lower energy cut-off was 5.6 MeV.

The authors should estimate and give the systematic uncertainty of the measured cross section because only the statistical uncertainty is given in Fig. 3.

Over the course of the experiment, three different CH_2 foil thicknesses were used that contributed to our overall normalisation and therefore were a possible source of systematic uncertainty. By using a variety of sources and staggering the energy each foil was used at, the systematic uncertainty contribution from the thickness uncertainty was dramatically reduced. Foil 1 was used for neutron energies of 8.36, 8.75, 9.25, 9.75 and 10 MeV. Foil 2 was used for neutron energies of 8.36, 8.6, 8.875 and 9.125 MeV. Foil 3 was used for neutron energies of 9.0 and 9.5 MeV. This way, the 3% systematic uncertainty in thickness is reduced to under 2% for the global magnitude of the cross section. The 3% error is included in the errors for each energy to account for this uncertainty on a point by point case.

An additional component of the systematic error is the uncertainty on the $^1\text{H}(n, p)$ (differential) cross section which is typically less than 1%.

Minor remarks: 1. The typical neutron flux on the ATTPC and the dimensions of the ATTPC should be given for readers' information.

Figure 2: (a) GEANT4 simulated excitation energies in ^{12}C (at $E_n = 10$ MeV) for the three reaction channels when passed through the analysis process. The separation between the three channels solely by the excitation is clear to see for events where the RANSAC garnered a good fit. (b) Channel separation from data for $E_n = 10$ MeV showing good separation between $^{12}\text{C}(n, n_2)$ in blue and $^{12}\text{C}(n, \alpha_{1,2})$ in red via the excitation energy in ^{12}C reconstructed from the three measured α -particles.

Figure 3: Simulated data for $E_n = 10$ MeV showing a histogram of the dot product between the longest and second-longest tracks against the length of the longest track showing clear channel separation. The $^{12}\text{C}(n, n_2)$ is blue, the $^{12}\text{C}(n, \alpha_1)$ is red and the $^{12}\text{C}(n, \alpha_2)$ is green.

Figure 4: NE-213 logarithmic spectrum across a broader energy range.

Figure 5: Rate enhancement as calculated as in the paper (solid line) and truncating the energy summation at 2 MeV in the COM (dashed) to match only our measured energy range. At 10 GK, the deviation is around 10%.

The neutron flux of 5000 n/s and the dimensions for TexAT have been included in the manuscript (lines 86 and 98).

In the legend of Fig. 3(a), “ $^{12}\text{C}(n,\alpha)3\alpha$ ” should be corrected as “ $^{12}\text{C}(n,n2)3\alpha$ ”. “ $^{12}\text{C}(n,\alpha)$ ” should be “ $^{12}\text{C}(n,\alpha_0)9\text{Be}$ ” for consistency with Ref. [5] and for readers to understand.

This change has been made.

It is better to show the neutron-energy range which is important to estimate the enhancement factors at various temperatures. If it is wider than that shown in Figs. 5, 6, 7, and 8, the authors should show the results from the R-matrix analysis in the wider energy range in those figures. If it is not wider, the authors should present the important energy range by something like gray shades in the figures.

Such a simple distinction is not possible as the relevant energy range varies as a function of temperature. To demonstrate this effect, the summation of Eq. 1 in the manuscript was truncated at 2 MeV in the COM in Fig. 5 to demonstrate the sensitivity to the rate enhancement outside of the measured energy range where the cross section is assumed to be featureless and almost constant. Even when this cross section dives down to zero above 2 MeV (i.e. outside our covered energy range), the sensitivity is less than 10% all the way up to 10 GK. In reality, the cross section cannot of course dive straight down to zero outside our energy range so this 10% is an absolute maximal deviation in the minimum direction.

The horizontal axes in Figs. 5, 6, 7, and 8 should be given as “Excitation energy of ^{13}C ” in order not to confuse readers.

This change has been made.

I guess that the $1/2^+$ resonance at $E_x = 13.0$ MeV in ^{13}C was considered in the present R-matrix analysis but its parameters were not given in Table I. If any states other than $E_x = 13.28$, 13.57 , and 13.76 MeV in ^{13}C were considered in the R-matrix analysis, all those states should be tabulated with their widths in Table I.

The position of this $1/2^+$ state has a very large uncertainty in the literature (up to 1 MeV) and in this work was identified as sitting at $E_x = 13.28$ MeV.

The caption of Table I should be revised as “Excited states in ^{13}C include in the ...”.

This change has been made.

2 Response to referee #2

This paper deals with the features of one component of the triple-alpha reaction, but they have done a good job of making a new measurement and then working out the consequences for the 3 α reaction. The work all appears to be carefully and competently performed, though without the experimental data and apparatus details I cannot vouch for the numerical accuracy of the result. My slightest correction is that the first legend in figure 3(a) should have n₂, not alpha₂. That corrected, I am much in favour of it being published.

The correction to Figure 3a has been made.

3 Response to referee #3

Key results: The rate, how fast carbon is formed in the centre of the stars, is one of the most important questions of nuclear astrophysics. Since the initial development of the concept in the early 1950s, the nuclear decays of the Hoyle state in C-12 at 7.6 MeV have attracted significant attention. In the laboratory, the Hoyle state dominantly decays by alpha particle into Be-8. However, a small portion of the time, C-12 will instead disintegrate into three alpha particles. Stable carbon is formed through the emission of an E2 transition, leading to the first excited 2+ state, or by an electric monopole transition, decaying directly to the ground state of C-12. In an astrophysical environment, alternative decay processes involving upscattering of protons, alpha particles or neutrons can enhance the overall carbon production. This paper is the first detailed experimental study to determine the enhancement of the triple-alpha reaction rate due to neutron-upscattering. Accurate determination of the total reaction rate has a large impact on our understanding of nucleosynthesis.

Methodology and validity:

The experiments have been carried out using highly collimated, quasi-monoenergetic neutron beams, covering the 8 to 10 MeV energy range, which is most relevant to astrophysical scenarios. A comprehensive experimental setup was used to monitor the beam and detect three alpha particles from the decay of the Hoyle state. A key component of the setup was the Texas Active Target Time Projection Chamber. Care was taken to calibrate the spectrometers, allowing the precise reconstruction of events. Extensive Monte Carlo simulations have also been used. A large data set was collected for inelastic neutron-scattering reactions, where the outgoing particle was an alpha particle or neutrons leading the final nuclear systems in excited or ground states. The neutron-upscattering reaction rates have been extracted from a detailed balance of the measured ‘normal’ reaction rates. The experimental rates are compared with multi-channel R-matrix calculations. Details of the measured rates are given in Figures 5 to 8 in the Appendix. These figures demonstrate the high quality of the data. To construct the plot of the enhancement of the radiative width of the Hoyle state due to neutron upscattering (Figure 4), the authors followed the procedure presented in ref [3] (Beard, Austin and Cyburt, 2017). This method was used by Davids and Boner (ApJ 166 [1971] 405) to study enhancement of the triple-alpha reaction by proton upscattering. The original formulation of the method was described by Fowler, Cauglan and Zimmerman (Ann. Rev. A.A. 5 [1967] 525). Like these earlier studies, the present paper describes the enhancement rate R according to equation (1) on page 4 of the manuscript. The detailed definition of the enhancement is not given in the paper, but can be obtained from ref [3]: $R = \tau_{\text{gamma}} / \tau_{n'n} = \tau_{\text{gamma}} * N_n * \langle \sigma v \rangle_{n'n}$, where τ_{gamma} is the radiative mean life of the Hoyle state, $\tau_{n'n}$ is the mean life of the neutron-upscattering channel, N_n is the neutron density and $\langle \sigma v \rangle_{n'n}$ is the cross section of the neutron upscattering. According to this definition, to determine R , one needs to use the value of the radiative lifetime of the Hoyle state. However, no details are given in the present paper as to what value has been used (the reviewer assumes it is included in the k_n factor). The most recent values of the relevant quantities to determine the radiative width are given in the 2014 review by Freer and Fynbo, ref [1] of the manuscript. Using the adopted values

from [1], one can obtain $\tau_{\text{gamma}}=1.77(17)\text{E-13 s}$. It is very close to the value given in ref [3]: $\tau_{\text{gamma}}=1.710\text{E-13 s}$. It should be noted that in the past few years, new results have been published on the E2 (Kibedi et al., PRL 125 [2020] 182701) and E0 (Eriksen et al., PRC 102 [2020] 024320) branching ratios, which change the value of τ_{gamma} by about 34%: $\tau_{\text{gamma}}=1.32(15)\text{E-13 s}$. Using the new value in determining the neutron-upscattering enhancement formula could further reduce the R values in Figure 4. Details of the determination of k_n , as well as the radiative width used in the calculations of Figure 4 should be included. This will make the results more robust, allowing it to be used in model calculations.

The k_n value used for the enhancement is based on the value used in M. Beard. As suggested, the revised value related to that suggested by Freer and Fynbo has been implemented. The most recent result by Kibedi is currently an experimental outlier in relation to the previous data so this most recent result has been omitted until further experimental validation. The E0 revised value is negligible in comparison to the dominant E2 strength. The manuscript has been modified to use this new value for k_n and explain the radiative width used:

“where k_n is a factor ($6.787 \times 10^{-6} \text{ K}^{\frac{3}{2}} \text{ cm}^3 \text{ g}^{-1}$ for inelastic neutron scattering for this system, using the radiative width suggested in ref. [1])”

This marginally affects the overall rate enhancement value therefore the numbers referred to in the text and the associated parameterisation are also modified accordingly.

Some minor comments and corrections: (a) Figure 2: It is not clear how the GEANT4 calculations were normalised to the experimental points. It seems the calculated curve underestimates experiments.

In Fig. 2, the GEANT4 curve has now been scaled to reproduce the experimental spectrum in the flat region between 8.2 and 8.36 MeV.

(b) Figure 3 caption: The data of Kuvin, ref. [9] is drawn from their Table III and not the figure. Ref [9] also has error bars, typically 10%, and should be shown in the figure.

Fig. 3 has had the errors bars from Table. III added.

Summary: Overall, it is a very impressive study, and, in my view, it should be accepted for publication in Nature Communications. However, I feel, that details of the parameterisation of R-ratio, outlined above, should be included in a minor revision of the paper.

4 Misc. comments

We have also added a section to the end of the manuscript detailed “Data Access” in order to clarify to readers how best they can access the data used in this work. Due to the nature of TPC data, the data files for this project total > 4 TB therefore require bespoke transferring of the data to any interested party upon written request.

References

- [1] E. Koshchiy, G.V. Rogachev, E. Pollacco, S. Ahn, E. Uberseder, J. Hooker, J. Bishop, E. Aboud, M. Barbui, V.Z. Goldberg, C. Hunt, H. Jayatissa, C. Magana, R. O’Dwyer, B.T. Roeder, A. Saastamoinen, and S. Upadhyayula. Texas active target (textat) detector for experiments with rare isotope beams. *Nuclear Instruments and Methods in Physics Research Section A: Accelerators, Spectrometers, Detectors and Associated Equipment*, 957:163398, 2020.
- [2] J. Bishop, G.V. Rogachev, S. Ahn, E. Aboud, M. Barbui, P. Baron, A. Bosh, E. Delagnes, J. Hooker, C. Hunt, H. Jayatissa, E. Koshchiy, R. Malecek, S.T. Marley, R. O’Dwyer, E.C. Pollacco, C. Pruitt, B.T. Roeder, A. Saastamoinen, L.G. Sobotka, and S. Upadhyayula. Beta-delayed charged-particle spectroscopy using textat. *Nuclear Instruments and Methods in Physics Research Section A: Accelerators, Spectrometers, Detectors and Associated Equipment*, 964:163773, 2020.
- [3] Neutron source capabilities at ohio university. <https://inpp.ohio.edu/~oual/facility/edwards-neutrons.pdf>. Accessed: 2022-Jan-11.

REVIEWERS' COMMENTS

Reviewer #1 (Remarks to the Author):

Most of my concerns in my previous report have been clarified. Now I agree to publish the present article in Nature Communications. I recommend that the authors describe about their tracking procedure with the RANSAC fitting technique in the supplement material. The explanation of the RANSAC fitting technique in the authors response is somewhat difficult for me to fully understand the procedure. The authors should explain it using figures.

Reviewer #3 (Remarks to the Author):

I am satisfied with the corrections and could recommend to accept the paper for publication.

Response to referees (round 2)

We would first like to once again thank all of the referees for their comments and suggestions on the manuscript. The referee comments are in bold red and our responses are in black.

1 Response to referee #1

Most of my concerns in my previous report have been clarified. Now I agree to publish the present article in Nature Communications. I recommend that the authors describe about their tracking procedure with the RANSAC fitting technique in the supplement material. The explanation of the RANSAC fitting technique in the authors response is somewhat difficult for me to fully understand the procedure. The authors should explain it using figures.

A figure detailing the idea behind RANSAC has been added to the Supplementary Figures (Supplementary Figure 2), shown below as Fig. 1.

2 Response to referee #3

I am satisfied with the corrections and could recommend to accept the paper for publication. We thank referee #3 for their comments.

Figure 1: Schematic showing the methodology of the RANSAC fitting routine. Three randomly-selected points for each arm (red, blue, green) for each iteration are made and 3D line segments are made between these three points and the vertex (purple). For each point in the 3D point cloud, the perpendicular distance to each of the three lines is calculated. For events where the distance between the point and the decay vertex is larger than the distance between the decay vertex and the arm (e.g. the final two points for the red arm), the 3D distance between the point and the arm point is used instead. The shortest of these three distances for each point is found and summed to provide a total distance-squared for each iteration. The random sample with the smallest total distance-squared is chosen as the best fit. This naturally finds not only the direction of the arms, but selects points at the end of the arm without over-biasing.